# Progress in and Prospects of Genome Editing Tools for Human Disease Model Development and Therapeutic Applications

**DOI:** 10.3390/genes14020483

**Published:** 2023-02-14

**Authors:** Hong Thi Lam Phan, Kyoungmi Kim, Ho Lee, Je Kyung Seong

**Affiliations:** 1Department of Physiology, Korea University College of Medicine, Seoul 02841, Republic of Korea; 2Department of Biomedical Sciences, Korea University College of Medicine, Seoul 02841, Republic of Korea; 3Graduate School of Cancer Science and Policy, National Cancer Center, Goyang 10408, Republic of Korea; 4Korea Mouse Phenotyping Center, Seoul National University, Seoul 08826, Republic of Korea; 5Laboratory of Developmental Biology and Genomics, BK21 PLUS Program for Creative Veterinary Science Research, Research Institute for Veterinary Science, College of Veterinary Medicine, Seoul National University, Seoul 08826, Republic of Korea; 6Interdisciplinary Program for Bioinformatics, Program for Cancer Biology, BIO-MAX/N-Bio Institute, Seoul National University, Seoul 08826, Republic of Korea

**Keywords:** programmable nuclease, genome editing, mouse model of human disease, therapeutic application

## Abstract

Programmable nucleases, such as zinc finger nucleases (ZFNs), transcription activator-like effector nucleases (TALENs), and clustered regularly interspaced short palindromic repeats (CRISPR)/Cas, are widely accepted because of their diversity and enormous potential for targeted genomic modifications in eukaryotes and other animals. Moreover, rapid advances in genome editing tools have accelerated the ability to produce various genetically modified animal models for studying human diseases. Given the advances in gene editing tools, these animal models are gradually evolving toward mimicking human diseases through the introduction of human pathogenic mutations in their genome rather than the conventional gene knockout. In the present review, we summarize the current progress in and discuss the prospects for developing mouse models of human diseases and their therapeutic applications based on advances in the study of programmable nucleases.

## 1. Introduction

Animal models are invaluable for unraveling pathogenic mechanisms and developing new therapeutics [1,2]. Given the long-standing efforts to generate human disease models for research, various mouse models, including knockout, knockin, humanization, point mutation, and transgenic models, have been produced with the assistance of embryo manipulation methods. Endogenous mouse genes can be modified or exogenous DNA may be inserted in the mouse genome through various manipulation tools. These tools include somatic cell nuclear transfer, microinjection of exogenous DNA into the pronucleus of a fertilized oocyte, microinjection of genetically modified stem cells into a blastocyst, and intracytoplasmic sperm injection (ICSI). The first genetically engineered animal was produced by injecting the Simian Virus 40 (SV40) DNA into mouse embryos in 1974, which ushered in a new era of transgenic animal models [3].

The emergence of programmable nucleases, such as zinc finger nucleases (ZFNs), transcription activator-like effector nucleases (TALENs), and clustered regularly interspaced short palindromic repeats (CRISPR)/Cas, has facilitated the production of genetically modified mouse (GEM) models with precise site-specific targeting abilities [4]. These programmable nucleases generate DNA double-strand breaks (DSBs) at the target sequences, which can be modified or repaired using DNA repair pathways, homology-directed repair (HDR), or non-homologous end joining (NHEJ). These repair processes result in insertion/deletion (indel) mutations, insertion of particular sequences, and correction of disease-associated variants at the target sequences. Recently, base editing (BE) and prime editing (PE) have been successfully developed and utilized to introduce more diverse mutations into the target sites [5,6]. When mouse models are generated using programmable nucleases, the conventional methods used are cytoplasmic or pronuclear microinjection or electroporation.

However, despite the ample studies describing the applications of engineered nucleases for developing mouse models of human diseases, comprehensive knowledge of the best practices for effective generation of mouse models as well as therapeutics based on nucleases need to be improved further. In the present review, we summarize the current progress in and prospects for developing mouse models using programmable nuclease-mediated gene editing technologies and their therapeutic applications.

## 2. Programmable Nucleases

Since the early-phase studies on genome editing, ZFNs and TALENs have been used to generate DSBs at the DNA target sites [7,8], and their structural features and editing mechanisms have been reviewed extensively [7,8,9,10]. Briefly, ZFN consists of a nonspecific cleavage domain of the *Flavobacterium okeanokoites*-derived FokI and a DNA-binding zinc-finger domain. About 3–6 tandem Cys2-His2 zinc fingers constitute a zinc-finger domain, each of which binds the corresponding three nucleotides [7,9,10]. Because of this hybrid structure, ZFNs are versatile and specific for DNA binding and robust cleavage. Therefore, ZFNs have emerged as potent tools for genome editing [9,10]. Similar to ZFNs, TALENs also exploit the FokI endonucleases for DNA cleavage. However, the DNA-binding domains in TALENs are transcription activator-like effector (TALE) repeats derived from the plant pathogenic *Xanthomonas* bacteria [8,11]. Compared to ZFNs, the construct of TALENs is less complex, and their application as molecular scissors is more specific and practical [12]. Even with this versatility and robustness, ZFNs and TALENs have several limitations, such as the need for reassembly of the ZFP or TALE modules for each target sequence. In addition, the activity of ZFNs is greatly dependent on the interaction specificity of adjacent fingers, which makes their overall function unpredictable, impeding their broad application [4,7,9,10]. Compared to ZFNs and TALENs, the CRISPR/Cas system containing a Cas nuclease and a single-guide RNA (sgRNA) can be used to modify target DNA sequences in a simpler manner [13,14]. Although this system may cause unwanted off-target effects, these can be minimized by selecting the sgRNA to have more specificity for the target sequence [15,16]. Of the Cas nucleases, the Cas9 nuclease has been commonly used as a genome editing tool, whose protospacer adjacent motif (PAM) sequence is 5′-NGG-3′. Another nuclease, Cpf1, known as Cas12a, uses a T-rich PAM, such as 5′-TTTV-3′, and exhibits various advantages over the Cas9 nuclease, including more frequent PAM sequences in the genome, smaller nuclease size, and lower incidence of off-target effects. Therefore, Cpf1 is an alternative to Cas9 nuclease and is beneficial for introducing knockin mutations when PAM sequences near the target site are less efficient [17]. Among the Cpf1 orthologs, AsCpf1, FnCpf1, and LbCpf1 nucleases have been frequently and widely utilized in gene editing [18,19,20,21].

CRISPR/Cas13, a recently developed RNA guide platform, recognizes single-stranded RNA instead of the double-stranded DNA [17,22,23,24,25]. The Cas13 enzyme has two higher eukaryote and prokaryote nucleotide-binding nuclease domains (HEPN), which degrade both the target and non-target RNAs randomly [22,23]. Mainly, Cas13 nuclease is more versatile than Cas9 or Cpf1, because it does not require PAM at the target site. In addition, RfxCas13d is the smallest protein that shows the most effective activity for RNA knockdown, regulation of alternative splicing, and viral resistance among the Cas13 orthologs [23,26,27,28,29].

Because of this facilitation and high applicability, genome editing tools have great potential for correcting mutations in human genetic disorders. However, their applications are limited owing to the generation of DSBs, which can cause indiscriminate indels at the target sites. Recently, the discovery of BE and PE has opened new opportunities to alleviate this limitation and hurdle because they do not generate DSBs or require donor DNA for gene corrections [30]. These editors have been demonstrated to be more efficient with lower off-target effects in vitro and in vivo than CRISPR/Cas9 [30,31]. Two BEs, cytosine BE (CBE) and adenine BE (ABE), have been reported; CBE consists of a nickase Cas9 (nCas9, D10A) or a dead Cas9 (dCas9, D10A, and H840A), a cytidine deaminase, and an uracil glycosylase inhibitor; ABE consists of nCas9 or dCas9 and evolved deoxyadenosine deaminases [31,32,33]. CBE replaces cytosine (C) with thymine (T) and ABE replaces adenine (A) with guanine (G) at the target site [31,32,33]. PE comprises a nickase Cas9 and a reverse transcriptase (RT); it has been known to be more versatile than CRISPR/Cas9, owing to its ability to generate various gene editing variants, including base transfers and small indels [6,31,33,34]. The recently developed BE and PE have been reported to have higher efficiencies and expanded target scopes in gene corrections [30,31].

Overall, owing to the straightforward, effective, and precise characteristics of the Cas-based systems, they are more extensively utilized for gene modifications [13,35]. These gene editing tools offer the opportunity to revolutionize the generation of mouse models with disease-associated variants.

## 3. Generation of Mouse Models of Human Diseases Using Genome Editing Tools

### 3.1. Mouse Models with Gene-Knockout or Point Mutations

Knockout strategies in mice using programmable nucleases have been reported in several studies [36,37,38,39,40,41,42,43,44,45,46,47,48,49,50,51,52]. Direct injection of the ZFN platforms into mouse embryos result in inactivation of the target genes, such as *Notch3*, *Mdr1a*, and *Jag1*, to generate GEM models [36]. Although ZFN-based methods have been effectively used to generate knockout mice [36,37,43], the complexity of target-specific ZFN manipulation and zygote toxicity have limited their extensive applications [37] (See Table 1 for Summary). 

The first reports of TALEN-mediated knockout mice involved *Pibf1* and *Sepw1* knockout mice that were generated by injecting TALEN mRNA into mouse embryos [38], in which about 49–77% of the pups had the mutation. In a subsequent study, the microinjection of TALEN mRNAs into mouse zygotes resulted in the knockout of several miRNAs, such as *mmu-mir-10a* and *-10b* [39]. In addition, another study found that applying two TALEN pairs targeting the FATS functional domain of the CFS gene resulted in larger deletions in size and bi-allelic mutations as well as further increase in the mutation efficacy compared to a single TALEN pair [40]. The cytoplasmic microinjection of *Ttc36*-targeted TALEN improved the knockout efficiency approximately three-fold compared to its injection into the pronuclei of mice [41]. The microinjection of TALEN mRNAs into the oocytes from CD1, C3H, and C57BL/6J mouse strains efficiently produced *Zic2* knockout mice [45]. These findings indicate that TALEN-mediated gene targeting is more effective in generating knockout mice than ZFNs because of the enhanced survival of the pups and higher efficiencies of introducing mutations.

It is well known that the CRISPR/Cas9 system is a powerful tool for producing GEMs in a highly efficient, simple, versatile, and cost-effective manner. Since the advent of the CRISPR/Cas9-mediated mutant mice, this approach has wholly replaced several platforms for genome manipulations that have been used commonly for over 30 years. CRISPR/Cas9-mediated enhanced green fluorescent protein (EGFP) knockout was achieved by co-injecting Cas9 mRNA and EGFP sgRNA into transgenic mice [53]. In addition, the cytoplasmic injection of *Tet1-* and *Tet2*-targeted Cas9 mRNA/sgRNA into mouse zygotes has been demonstrated as a reliable and efficient method for generating bi-allelic mutations [42]. The cytoplasmic microinjection of Cas9 mRNA and sgRNA has been reported as the most efficient method for gene knockout in mice as opposed to pronuclear microinjection [52,54]. The microinjection of Cas9 mRNA and sgRNA into the pronuclei of mice following in vitro fertilization also generated indel mutations in the *Fah* allele in NOD-*Rag1^null^IL2Rγ^null^* mice [55]. The cytoplasmic injection of Cas9 and truncated sgRNA complexes were reported to disrupt FVII with high efficiency compared to standard complexes composed of Cas9 and standard sgRNA [44]. CRISPR/Cas9 has been utilized to disrupt the *ATP6V1H* gene, which causes bone loss in mice [56]. Deleting the *Notch3* gene via the injection of Cas9/sgRNA into the mice embryos was reported to generate a human lateral meningocele syndrome-associated mouse model, which displayed a reduction in several femur trabecular scores, such as the cancellous bone volume, connectivity, and trabecular number [57]. Pankowicz et al. reported a knockout mouse for the liver, in which the CRISPR/Cas9 system deleted both *Fah* and *Hpd* or *Hpd* and *Asl* genes. These mice could be used to develop liver-disease-related models as well as further investigate metabolic liver disorders and liver function-related genes [58,59]. A previous study rapidly and effectively generated a mouse model with a 5 Mb deletion of the *SOX9* gene by microinjection of Cas9/sgRNAs and a donor vector into fertilized eggs [60]. Similarly, various studies have reported the deletions of multiple genes, such as the 0.5 Mb (*Irx3* and *Irx5* genes), 1.15 Mb (intervals from *Nox4* to *Grm5* genes, including *Tyr* locus), 1.2 Mb (regions carrying nine genes, including *Srd5a3*, *Exoc1*, *Nmu*, *Kit*, *Kdr*, *Clock*, *Cep135*, *Tmeme165*, and *Pdcl2*), 30 kb (*Pcdh*), and 95 kb (*Fpr1-3*). However, the efficiencies of these large deletions were relatively low [51,61,62,63,64]. A previous study reported acampomelic campomelic dysplasia (ACD) and campomelic dysplasia (CD) mouse models in which the CRISPR/Cas9 system was used to knock out the *SOX9* gene, a transcription factor involved in the development of cartilage [65]. Furthermore, adeno-associated viral (AAV)-expressing sgRNA for *p53*, *LKB1*, and *KRAS* genes was delivered to the lung of mice expressing CRISPR-Cas9 via tail vein injection, resulting in target gene mutations and lung tumorigenesis [66].

Since the development of base editing technologies capable of editing specific nucleotides, the CBE method has been used to generate *Tyr* or *Dmd* mutants with C to T conversion by delivering sgRNA and CBE mRNA or ribonucleoprotein (RNP) to mouse zygotes [67]. Subsequently, the CBE-based CRISPR-stop method has been used to generate mutant mouse models with stop codons by converting single nucleotides in the coding sequences of various genes, such as *Tyr*, *vGlut3*, *Slc26a5*, *Otof*, and *Atoh1* [68]. Wang et al. reported the generation of a PD-1 mutant mouse using a novel strategy known as induce gene silencing (i-Silence), with an ABE-mediated start codon mutation from ATG to GTG or ACG [69]. Using the PE system, a mouse model has been generated to have one deletion in exon 3 of the *Crygc* gene by microinjection of the PE3 plasmid directly into the mouse zygote that shows a cataract phenotype [70]. These findings provide a promising foundation that such editing tools can be extensively used for editing or correcting human disease-associated mutations.

**Table 1 genes-14-00483-t001:** Generation of mouse models using gene editing tools. NA: not available, without information of related diseases in general.

Gene Editing Tool	Target Gene	Human Disease Models	Mutation Type	Manipulation Methods	Reference
ZFN	*Notch3*, *Mdr1a*, *and Jag1*	NA	Knockout	Microinjection	[36]
TALEN	*Pibf1 and Sepw1*	NA	Knockout	Microinjection	[38]
TALEN	*mmu-mir-10a and -10b*	NA	Knockout	Microinjection	[39]
TALEN	*Ttc36*	NA	Knockout	Microinjection	[41]
TALEN	*Zic2*	NA	Knockout	Microinjection	[45]
CRISPR/Cas9	*Tet1 and Tet2*	NA	Knockout	Microinjection	[42]
CRISPR/Cas9	*Fah*	Tyrosinaemia	Knockout	Microinjection	[55]
CRISPR/Cas9	*FVII*	Hemophilia	Knockout	Microinjection	[44]
CRISPR/Cas9	*ATP6V1H*	Osteoporosis/ osteopenia	Knockout	Microinjection	[56]
CRISPR/Cas9	*Notch3*	Lateral meningocele syndrome	Knockout	Microinjection	[57]
CRISPR/Cas9	*Fah and Hpd*	Tyrosinaemia type I	Knockout	Hydrodynamic injection	[58]
CRISPR/Cas9	*Hpd and Asl*	Hyperammonemia	Knockout	Hydrodynamic injection	[59]
CRISPR/Cas9	*SOX9*	Acampomelic campomelic dysplasia (ACD)	Knockout	Microinjection	[65]
CRISPR/Cas9	*p53*, *LKB1*, *and KRAS*	Lung adenocarcinoma	Knockout	Microinjection	[66]
CBEs	*Dmd*	NA	Knockout	Microinjection/Electroporation	[67]
CBEs	*Tyr*	Pigmentation disorder	Knockout	Microinjection/Electroporation	[67]
CBEs	*Atoh1*, *vGlut3*, *Slc26a5*, *Otof*	Hearing loss	Knockout	Microinjection	[68]
ABEs	*PD-1*	NA	Knockout	Microinjection	[69]
PE	*Crygc*	Cataract disorder	Knockout	Microinjection	[70]
ZFN	*Rab38*	NA	Knockin	Microinjection	[71]
TALEN	*Rab38*	NA	Knockin	Microinjection	[72]
TALEN	*Fus*	Amyotrophic lateral sclerosis	Knockin	Microinjection	[73]
TALEN	*Crb1*	Retinal disease	Knockin	Microinjection	[74]
TALEN	*Scn8a*	Epileptic encephalopathy	Knockin	Microinjection	[75]
TALEN	*Oct4 and Nanog*	NA	Knockin	Microinjection	[76]
CRISPR/Cas9	*Actb*	NA	Knockin	Microinjection	[77]
CRISPR/Cas9	*BRIL*	Osteogenesis imperfecta (OI) type V	Knockin	Microinjection	[78]
CRISPR/Cas9	*HTT*	Huntington’s disease (HD)	Knockin	Microinjection	[79]
ZFN	*CCR5*	HIV-1	Humanized	Transplantation/Engraftment	[80,81,82]
TALEN	*HBG*	β-thalassemia and sickle cell disease	Humanized	Transplantation/Engraftment	[83]
CRISPR/Cas9	*DMD*	*DMD*	Humanized	Microinjection	[84]
CRISPR/Cas9	*Pdcd1 and Lag3*	NA	Humanized	Electroporation	[85]

### 3.2. Knockin Mouse Models

Over the last ten years, the number of newly developed mouse models has increased dramatically, which has extended the applications of programmable nucleases. The first report of ZFNs for the generation of knockin models was for the insertion of the GFP-linked *β-Galactosidase* gene into the *Rosa26* locus with a 1.7–4.5% targeting efficiency [43]. Other knockin models have also been produced by inserting reporter cassettes into the *Rosa26* or *Mdr1a* locus using ZFNs [37,86]. It was reported that the injection of *Rab38*-targeted ZFNs or TALENs with a 144-base single-stranded oligodeoxynucleotide (ODN) into single-cell embryos resulted in homologous recombination (HR)-mediated modifications with approximately 2% efficiency [71,72]. In addition, the fusion of TALEN-95As, an improved plasmid-coded poly(A) targeting the *Fus* gene, and the 140-base mutant ODN has been used to produce an amyotrophic lateral sclerosis (ALS)-related mouse knockin model with increased knockin efficiency of up to 6.8% [73]. Microinjecting TALEN mRNAs into the pronucleus generated TALEN-mediated knockin mice for the target genes, such as *Satb1*, *Crb1*, and *Scn8a* [74,75,87]. The injection of EGFP/mCherry reporter-linked *Oct4* and *Nanog* vectors, Cas9 mRNA, and sgRNAs into either the cytoplasm or pronucleus resulted in effective insertion with 9–19% efficiency [76]. In addition, an injection experiment with a mixture containing an EGFP-linked *Actb* vector, Cas9 protein, *Actb* crRNA, and tracrRNA into the pronuclei showed increased insertion efficiency by 50%. These results suggest that this is one of the practical and convenient strategies for producing knockin mouse models [77]. Another knockin mouse model using CRISPR/Cas9 has been generated to induce *BRIL* variants found in patients with osteogenesis imperfecta type V [78].

The CAG repeat sequences in huntingtin (*HTT*) gene have been reported to cause Huntington’s disease (HD) [88]. Yang et al. generated a knockin HD mouse model with CAG repeat sequences in the *HTT* genes using CRISPR/Cas9. They evaluated the toxicity of the repeat-associated non-AUG-mediated products in the mouse model [79]. This development of *HTT* knockin mice using CRISPR/Cas9 is expected to be a promising breakthrough for researching pathogenesis and therapeutics for HD disease.

### 3.3. Humanized Mouse Models

Humanized models can be generated by transplanting short strands of DNA sequences, cells, tissues, or microbiomes from humans to mice. They are broadly exploited in the study of various human diseases, such as HD, human immunodeficiency virus type 1 (HIV-1), Charcot–Marie–Tooth type 1 (CMT1), and cancers [80,89,90,91]. The generation of humanized mouse models using gene editing technologies typically involve insertion of human-derived DNA sequences at specific sites or transplanting of gene-edited human-derived cells into mice.

Using *CCR5*-targeted ZFNs, the *CCR5* gene, a co-receptor playing a fundamental role in HIV-1, was disrupted in human hematopoietic stem/precursor cell (HSC) with a mutant efficiency of 17% [80,81,82]. Edited HSCs were transplanted in NOD/SCID/*IL2Rγ^null^* (NSG) mice, which can be a potential model for developing HIV-1 treatment because the deletion in the HSCs is maintained even after maturation [80,81,82]. β-thalassemia and sickle-cell disease are hemoglobin disorders caused by pathogenic mutations of the *HBG* gene. TALENs targeting the *HBG* promoter were employed to drive the hereditary persistence of the fetal hemoglobin phenotype (6%) in human CD34^+^ cells. The edited cells were transplanted in W41 mice and were investigated for persistence in hematopoietic cells for 24 weeks [83]. Duchenne muscular dystrophy (DMD) is a muscular disorder caused by mutations in the dystrophin gene on the X chromosome [92]. A Δ50;h51KI humanized DMD mouse model was generated using CRISPR/Cas9 targeting the *DMD* gene; in these mice, exon 51 was replaced with human exon 51 and exon 50 was deleted [84]. To generate human *PD-1/LAG-3* knockin mice expressing human PD-1 protein, mouse *Pdcd1* and *Lag3* genes were replaced with human *Pdcd1* and *Lag3* genes using CRISPR/Cas9 [85]. This dual-immune checkpoint model is expected to enable evaluation of human immune responses, including the analyses of cytokine release, tumor growth, and lymphocyte populations.

### 3.4. Transgenic Mouse Models

Transgenic mice can be generated by microinjecting exogenous recombinant DNA directly into the pronuclei of fertilized oocytes. The recombinant DNA is randomly inserted in the chromosome and expressed by an exogenous promoter [93]. However, the practical application of this method is limited as the expression of the transgene is critically affected by the integration site in the chromosome [94].

Recently, CRISPR/Cas9-based conditional transgenic mice harboring Rosa26-LSL-dCas9-p300 or Rosa26-LSL-dCas9-KRAB have been generated, which express transgenes in a Cre-dependent manner. The Rosa26-LSL-dCas9-p300 mice were designed for gene activation and Rosa26-LSL-dCas9-KRAB for gene repression. The target genes were regulated by targeting gRNAs to the transcriptional start sites or distal enhancer elements in the transgenic mice, resulting in corresponding changes in the epigenetic states and downstream phenotypes in the brain and liver [95]. These transgenic mice could be applied to more diverse functional and mechanism studies using tissue-specific Cre and sgRNAs.

## 4. Therapeutic Applications Using Genome Editing Tools

### 4.1. Duchenne Muscular Dystrophy (DMD)

DMD, an X-linked recessive disorder, is caused by a mutation in the *DMD* gene, which impairs normal dystrophin function. The mouse model with a point mutation in the *DMD* gene is the *mdx* mouse, which is the most popular model for studying DMD [96,97,98,99,100]. Several recent studies have attempted to restore dystrophin function by eliminating *DMD* mutations using programmable nucleases [96,97,98,99,100]. Compared to ZFNs and TALENs, the CRISPR/Cas9 system has been applied more commonly in studies on treating DMD. Several studies have reported the AAV-mediated CRISPR/Cas9 platform for correcting *DMD* mutations in dystrophic *mdx* mice and observed significantly improved muscular functions in the treated mice [96,97,98,99,100]. In addition, CRISPR/Cas9 has been utilized to generate a novel DMD mouse model with the disruption of exon 50 of the *DMD* gene, which exhibits severe muscle dysfunction. Treatment with AAV9 expressing both Cas9 and exon-51-targeted sgRNA in the mouse resulted in restoration of dystrophin expressions in the cardiac and skeletal muscles by up to 90% [101]. The RNA-guided endonuclease CRISPR/Cpf1 has been used to correct a nonsense mutation in exon 23 in *mdx* mice, which was efficiently and permanently restored by injecting LbCpf1 mRNA, 180-base ssODN, and sgRNA into the mouse zygotes [19]. Various follow-up studies have reported that AAV9-mediated CRISPR/Cas9 delivery effectively restored dystrophin expression in DMD mouse models harboring deletions of exons 43, 44, 45, 51, or 52 [84,102,103,104]. Another study showed that these gene corrections and dystrophin protein expressions by AAV-CRISPR treatment were maintained in *mdx* mice for up to one year [105]. Although the development of AAV-CRISPR as a therapeutic method must consider unwanted gene editing and transcriptome modifications, it is undeniable that AAV-CRISPR may offer the potential to correct the genome permanently [105] (See Table 2 for Summary). 

Ryu et al. reported that a *DMD*-gene-targeted ABE7.10 base editor was delivered into mouse muscle using double trans-splicing AAV and that up to 17% of exon 20 mutations in the DMD mice were corrected without off-target effects [106]. However, the large size of ABE7.10 with sgRNA (~6.1 kb) limits the common use of these BEs [106]. To address this limitation, iABE-NGA, a mini adenine BE targeting NG, was developed. Delivery of the AAV-packaged iABE-NGA into *mdx^4cv^* mice precisely and effectively corrected *DMD* mutations and recovered dystrophin expression without toxicity and with low off-target effects [107]. In addition, the delivery of CRISPR/Cas9 RNP complexes by nanoparticles, such as 5A2-DOT-X LNPs, into the tibialis anterior muscles of the exon-44-deleted DMD mice restored dystrophin expression by 4.2% [108]. Recently, CRISPR/Cas9 was used to correct induced myogenic progenitor cells (iMPCs) derived from *DMD^mdx^* mice with mutations in exon 23. The corrected iMPCs were subsequently engrafted into dystrophic limb muscles to successfully restore dystrophin protein expression [109]. Overall, these approaches for gene correction in DMD models might provide promising strategies for repairing pathogenic mutations in patients with muscular dystrophies.

**Table 2 genes-14-00483-t002:** Gene editing-mediated therapeutics using mouse models of human disease.

Human Disease	Mutant Gene	Mutation Type	Gene Editing Tool	Delivery Method	Reference
Duchenne muscular dystrophy (DMD)	*DMD*	Knockout	CRISPR/Cas9, CRISPR/ Cpf1, and ABE	AAV/RNP	[105,106,107,108,109]
Hemophilia A	*F8*	Knockout	CRISPR/Cas9	AAV	[110,111,112]
Hemophilia B	*F9*	Knockout	ZFN and CRISPR/Cas9	AAV	[113,114,115,116,117,118]
Huntington’s disease (HD)	*HTT*	Humanized	ZFNs, CRISPR/Cas9, CRISPR/Cpf1, and RfxCas13d	AAV	[18,90,119,120,121,122,123]
Amyotrophic lateral sclerosis (ALS)	*SOD1*	Transgenic	CRISPR/Cas9, RfxCas13d, and CBE	AAV	[124,125,126,127]
Hearing loss	*TMC1/* *Kcnq4*	Knockout	CRISPR/Cas9, RfxCas13d, PE, and CBE	AAV/RNP	[128,129,130,131,132,133]
Hereditary tyrosinaemia type I (HT-I)	*Fah*	Knockout	CRISPR/Cas9, ABE, and PE	AAV/LNP/Plasmid	[58,134,135,136,137,138,139,140]
Neovascular age-related macular degeneration (AMD)	*Vegfa/* *HIF-1α*	Knockout	CRISPR/Cas9 and LbCpf1	AAV/RNP/ Lentivirus	[141,142,143,144]
Charcot-Marie-Tooth disease (CMT1)	*PMP22*	Humanized	CRISPR/Cas9	RNP	[89,145]
Leber congenital amaurosis (LCA)	*Rpe65*	Knockout	CRISPR/Cas9, ABE, and PE	AAV/Lentivirus/ eVLPs	[146,147,148,149]
Alzheimer’s disease (AD)	*Bace1/* *Mt1*	Transgenic	CRISPR/Cas9	RNP/Lentivirus	[150,151]
Familial hypercholesterolemia (FH)	*Ldlr*	Knockout	CRISPR/Cas9	AAV	[152]
Pompe disease (PD)	*Gaa*	Knockout	CRISPR/Cas9	Electroporation	[153,154]
Progeria	*Lmna*	Knockout	CRISPR/Cas9	AAV	[155,156]

### 4.2. Hemophilia

Hemophilia is an X-linked hemorrhagic disorder characterized by the deficiency of clotting factors VIII (F8, Hemophilia A) or IX (F9, Hemophilia B) [157]. Injecting the *Alb* intron-13-targeted AAV8-SaCas9-gRNA and AAV8-modified human B domain deleted-F8 into FVIII hemophilic knockout mice resulted in elevation of their plasma FVIII levels and amelioration of blood loss without liver toxicity and off-target effects [110]. Other studies have reported the rescue of F8 expression in hemophilia A using CRISPR/Cas9 and AAV delivery [111,112]. Anguela et al. found that the injection of AAV-ZFN expressing *hF9* gene and AAV-donor harboring homologous arms to the target gene into *hF9* mutant mice resulted in an approximately five-fold increase in the expression of hF9 protein compared to a previous study [113,114]. A hemophilia B mouse model carrying the Y371D mutation in the *F9* gene was previously generated using the CRISPR/Cas9 system [115]. Mutant mice were used for gene correction experiments in which Cas9 and *F9*-targeted sgRNA components were delivered via adenoviral (AdV) or naked DNA vectors. Direct injection of the naked DNA resulted in correction of more than 0.56% of the *F9* mutations, which are sufficient for restoring the hemostasis phenotype. In contrast, the AdV delivery showed no therapeutic application because of severe toxicity in the hepatocytes of the mice, despite the remarkable repair efficacy [115]. Another study demonstrated that the AAV-mediated delivery of CRISPR/Cas9 to the hemophilia B mouse model recovered FIX expression [116,117,118]. Recently, the injection of dual AAV-CjCas9 targeting *hF9* and a donor template into the hemophilia mice induced hFIX expressions and improved coagulation functions [116]. Together, these experimental data strongly support the observation that genome editing tools can be used therapeutically in the clinical treatment of hemophilia.

### 4.3. Huntington’s Disease

Huntington’s disease (HD), one of the neurodegenerative disorders, is characterized by the expansion of CAG repeats that encode polyglutamine in the *HTT* gene [88]. Isalan et al. reported that the delivery of a striatal AAV vector harboring zinc fingers (ZFs) targeting pathogenic CAG-repeat expansions into the R6/2 mice, an HD mouse model, resulted in the suppression of cerebral HTT expression and reduction of HD-associated symptoms [119]. In addition to ZFs, AAV-mediated deliveries of CRISPR/Cas9 and CRISPR/Cpf1 have been ubiquitously utilized to remedy HD via *HTT* gene editing [18,120,121,122,123]. Delivery of AAV1 packaging RfxCas13d, a Cas13 variant targeting *hHTT* exon 1 in R6/2 HD mice, decreased HTT expression by approximately 50%, suggesting the ability of RfxCas13d to ameliorate HD-linked pathology [124]. This finding indicates that RfxCas13d may be a potential tool for suppressing *HTT* and neurodegeneration-related genes in the central nervous system.

### 4.4. Amyotrophic Lateral Sclerosis

Amyotrophic lateral sclerosis (ALS) is a neurodegenerative disorder characterized by the malfunction of nerve cells in the brain [158]. *SOD1* mutations are the most prevalent and fundamental cause of ALS [159]. Because CRISPR/Cas effectively interferes with the expressions of the mutant genes in *SOD1* ALS mouse models, this editing technology has been considered a potential therapeutic tool for ALS [124,125,126,127]. Duan et al. reported that CRISPR/Cas9 targeting *SOD1^G93A^* mutations was delivered by AAV9 vehicles to ALS transgenic mice, disrupting the mutant *SOD1* gene and increasing the lifetime of the mice by up to 54.6%. Furthermore, the ALS phenotype of the mice, including muscle atrophy and paralysis, was improved [126]. The AAV9-packaged RfxCas13d targeting mutant *SOD1* reduced 50% of the expression of SOD1 in the ALS spinal cord, followed by functional improvement of neuromuscular disorders as well as decrease in muscle atrophy [124]. Interestingly, dual-AAVs-delivered split-intein cytidine base editors reduced the expression of mutant *SOD1*, leading to the amelioration of muscle denervation and muscle atrophy and improvement of neuromuscular function in ALS mice [127]. Together, these CRISPR-mediated therapeutic applications for ALS have great therapeutic potential because they have become more sophisticated and diverse.

### 4.5. Hearing Loss

Hearing loss is one of the most prominent disorders among newborns and is caused mainly by mutations in the protein transmembrane channel-like 1 (*Tmc1*) gene [160]. Several studies have reported gene therapy strategies targeting *Tmc1* using CRISPR/Cas9 or CBE [128,129,130]. Mutant *Tmc1*-targeting CRISPR/Rfx Cas13d was packaged in AAV-PHP.eB, a previously developed novel AAV vector system [131]. Gene editing with this CRISPR/Cas system reduced the expression of *Tmc1* by up to 70.2% in the cochlea of *Bth* neonatal mice, resulting in alleviation of the pathological symptoms in the mice [132].

Degeneration of the outer hair cells (OHCs) has also been reported to cause hearing loss [161]. Noh et al. generated a *Kcnq4* mutant mouse exhibiting an OHC degenerative phenotype. When dual AAV vectors or RNP harboring SpCas9 and sgRNA were injected, the *Kcnq4* mutant mouse showed partial recovery of hearing function [133]. These results are strong evidence that the therapeutic applications of CRISPR/Cas9 are promising alternatives for treating hearing loss.

### 4.6. Hereditary Tyrosinemia Type I

Hereditary tyrosinemia type I (HT-I) is an inherited metabolic disorder that results from a mutation in the *Fah* gene encoding fumarylacetoacetate hydrolase [162]. When Cas9 and *Fah*-targeting sgRNA were injected into *Fah5981SB* mice carrying a homozygous G to A variant located at the last base of exon 8 of the gene, the pathological phenotypes of the mice were improved [134]. VanLith et al. transduced a lentiviral vector harboring Cas9 nuclease and *Fah* locus-targeting sgRNA as well as an AAV particle carrying a 1.2 kb homologous template to the *Fah^−/−^* hepatocytes. When these hepatocytes were transplanted in *Fah^−/−^* mice, they found that the edited hepatocytes improved liver injuries and restored metabolic disorder [135]. Since then, several studies have demonstrated that editing the mutant *Fah* gene via the delivery of AAV-encoding CRISPR or PE components could help recover the lethal phenotype of the *Fah*-deficient mice [58,136,137]. Although successful recoveries of the pathogenic phenotypes were observed, the AAV system has critical limitations for common application. The AAV vector can allow DNA fragments of up to 4.7 kb because of its loading capacity. Recently, a promising alternative method has been reported: delivery of CRISPR/Cas9 components, including ABE mRNA and *Fah*-targeting sgRNA by lipid nanoparticles (LNPs) [138,139]. Jiang et al. reported that a PE-Cas9-based deletion and repair (known as PEDAR) method could recover the expression of functional FAH in a Tyrosinemia I mouse model by accurately eliminating a 1.38 kb pathogenic fragment of the *Fah* gene via liver-targeted hydrodynamic injection [140]. These results demonstrate the outstanding application of PEDAR in the repair of *Fah* in Tyrosinemia I mice and its potential for correcting pathogenic genes caused by insertion of large fragments.

### 4.7. Age-Related Macular Degeneration (AMD)

Neovascular AMD is one of the most prevalent causes of vision loss globally [163]. Several studies have adapted gene editing tools to modulate the vascular endothelial growth factor (*Vegfa*) and hypoxia-inducible factor-1α (*HIF-1α*) associated with AMD [141,142,143,144]. Co-delivery of two separate lentivirus particles harboring Cas9 mRNA and sgRNA to an AMD mouse model resulted in 44% deletion of *Vegfa* in the retinal pigment epithelial (RPE) cells and 63% decrease in choroidal neovascularization (CNV). Kim et al. exploited the subretinal injection of RNPs, composed of Cas9 and sgRNA, into an AMD mouse model to cause mutations in *Vegfa* [144]. Another study reported that gene editing with Cas9 RNPs successfully reduced CNV in AMD mice [143]. When the AAV-packaged LbCpf1 targeting *Vegfa* or *Hif1a* was intravitreally injected into laser-treated mouse retinas, CNV was effectively reduced without detectable cone damage. It has been reported that the LbCpf1 approach is more effective than Aflibercept, a VEGF inhibitor commonly used in clinical practice, for CNV reduction. Moreover, these effects of the LbCpf1 system lasted longer without repeated injections [142]. Collectively, these results suggest that gene editing tools with LbCpf1 could be potent therapeutic agents for treating AMD.

### 4.8. Charcot–Marie–Tooth Type 1

Charcot–Marie–Tooth type 1 (CMT1) is a disease involving various genetic abnormalities that characterize the phenotypes of motor and sensory neuropathies [164]. The duplication of peripheral myelin protein (*PMP22*) explains 70% of CMT1 cases [164]. Recently, CRISPR/Cas9 RNA complexes targeting the TATA-box of *hPMP22* were intraneurally injected into a C22 mouse, resulting in a decrease in PMP22 expression and reduction of demyelination [145]. This is a pioneering report indicating that the CRISPR/Cas9 system could be a promising therapeutic strategy for treating CMT1 disease.

### 4.9. Leber Congenital Amaurosis

Leber congenital amaurosis (LCA) is one of the leading causes of blindness in infants. Among LCA-linked genes, *RPE65* mutations account for approximately 6% of the LCA cases [165]. In a previous study, two separate AAV vectors expressing Cas9 and including a donor template linked to a TS4 sgRNA targeting a mutation in exon 3 of the *Rpe65* gene were delivered to rd12 mice, a naturally occurring model of LCA with a Rpe65 mutation, via subretinal injection. The treated LCA mice showed improved pathogenic phenotype by deleting approximately 1.6% and causing more than 1% HDR of the *Rpe65* mutation [146]. Subretinal injection of ABE- and sgRNA-expressing lentiviral particles in the rd12 mice efficiently corrected mutations in the *Rpe65* gene (29%) with low indels and no off-target effects. ABE treatment in the *Rpe65* mutant mice increased RPE65 expressions and restored visual functions, which were mediated by functional restoration and prolongation of survival of the cone photoreceptors [147,148]. Jang et al. recently generated an AAV vector encoding *Rpe65*-targeting pegRNA and PE2; subretinal injection of the vector into rd12 mice resulted in efficient editing of the target gene by up to 6.4% without any unwanted editing and revealed dramatic improvements to visual function [149]. Together, these results indicate that high-precision editing using PE2 and AAV delivery may be a potential option for treating patients with LCA.

### 4.10. Alzheimer’s Disease

Alzheimer’s disease (AD) is the most common neurodegenerative disorder characterized by slow memory destruction, synaptic impairment, and Aβ accumulation [166]. Since the first report of type 1 melatonin receptor (*Mt1*) and β-secretase 1 (*Bace1*) for the regulation of Aβ production, targeting *Mt1* and *Bace1* has been considered a powerful therapeutic approach for Alzheimer’s disease [167]. Park et al. generated amphiphilic nanocomplexes loaded with Cas9 and *Bace1*-targeting sgRNA and then injected into the hippocampi of *5XFAD* and *APP* transgenic AD mouse models. This approach effectively reduced *Bace1* expression and cognitive deficiencies to alleviate the Aβ-associated pathologies in the AD mouse models [150]. In addition, injection of an *Mt1*-Cas9 activator into the brain of the *5xFAD* mice efficiently activated Mt1 expression and relieved cognitive deficits in AD mice [151].

### 4.11. Familial Hypercholesterolemia

Familial hypercholesterolemia (FH) is an inherited disease characterized by significant elevation of the low-density lipoprotein cholesterol [168]. This disease is mainly caused by mutations in the *Ldlr* gene [169]. Zhao et al. developed CRISPR/Cas9-loaded AAV particles to correct these *Ldlr* mutations; the particles were transferred to the hepatocytes of a *Ldlr^E208X^* mouse model, which effectively reduced atherosclerosis phenotypes in the mutant mice [152]. Overall, this finding demonstrates that *Ldlr* editing using AAV-CRISPR/Cas9 can serve as a potential therapeutic strategy for treating FH.

### 4.12. Pompe Disease

Pompe disease (PD) is an autosomal recessive disease caused by mutations in the *GAA* gene, which encodes acid α-glucosidase, an essential enzyme that hydrolyzes lysosomal glycogen [170]. Through microinjection into fertilized zygotes using CRISPR/Cas9-mediated HDR, mice with infantile-onset PD were generated expressing *Gaa^c.1826dupA^* or *Gaa^Em1935C>A^*. The mice exhibited hypertrophic cardiomyopathy and skeletal muscle deficiency associated with PD [153,154]. This PD mouse model can be used for gene therapy research targeting *Gaa* mutations as well as mechanisms related to PD.

### 4.13. Progeria

Progeria, also known as Hutchinson–Gilford progeria syndrome (HGPS), is a rare genetic disorder characterized by premature aging in children, leading to early death [171]. De novo point variants in the *LMNA* gene are known to cause 90% of HGPS cases [172]. Fernández et al. and Beyret et al. delivered an AAV9-packaged CRISPR/Cas9 system targeting *LMNA* exon 11 intraperitoneal injection into *Lmna^G609G/G609G^* HGPS mice. Despite the low editing efficacy of this approach, it not only improved the HGPS-specific phenotypes but also extended the lifespans of the HGPS mice [155,156]. Although further studies are needed to evaluate the side effects and off-target effects of CRISPR/Cas9, this finding opens up the possibility of applying CRISPR/Cas9 to HGPS treatment.

## 5. Conclusion and Future Prospects

Recent technological advances in programmable nucleases, including ZFNs, TALENs, and CRISPR/Cas, have significantly contributed to the development of therapeutic strategies for various diseases in human. In particular, ZFNs, TALENs, and CRISPR/Cas9 have been applied in clinical trials [173,174]; among these three genome editing tools, CRISPR/Cas has been proven to regulate the expressions of target genes simply and shows promise for development into various versatile tools. Accordingly, novel miniature CRISPR/Cas12f nucleases, such as AsCas12f1, Un1Cas12f1, and SpaCas12f1 (422, 529, and 497 amino acids, respectively), have recently been indicated as effective editing tools in bacteria, cells, and tissues using delivery strategies, such as plasmids, RNPs, and AAV [175,176,177]. CRISPR-Casπ (Cas12I) is a recently developed miniature type V CRISPR/Cas system comprising 860 amino acids that has demonstrated significant editing capacity for cells compared to two well-engineered editors, namely LbCas12a and SpyCas9 [178]. A recent study reported Cas12f-based ABEs and CBEs with engineered TadA, which show successful gene editing with minimal off-target effects both in vitro and in vivo via the all-in-one AAV delivery system [179]. These mini gene editors can be easily packaged in an AAV vehicle to provide promising therapeutic options for various human diseases (See Figure 1 for Summary).

Although the off-target effects in a genome editing platform remain to be studied, sophisticated programmable nuclease-containing complexes and specialized nanoparticles have effectively improved the editing efficacies with minimal toxicity in mouse models. Specifically, lipid nanoparticles (LNPs), which are an FDA-approved approach for delivering siRNA to hepatocytes, have been developed and effectively exploited for sgRNAs and Cas9 mRNA delivery to mice, thereby allowing gene editing in not only the liver but also other organs [180,181]. Interestingly, Cas9 RNA- and sgRNA-packaged LNPs have been successfully used to treat patients suffering from transthyretin amyloidosis in an ongoing phase 1 clinical research [182]. A recent study demonstrated effective delivery of Cas9- or BE-RNPs by engineered DNA-free virus-like particles (eVLPs); this delivery method exhibits undetectable off-target effects and superiority compared to plasmid-, lentivirus-, or AAV-based delivery [183]. RNP-packaged eVLPs have been suggested as a favorable alternative to existing delivery approaches, owing to their transient exposure to gene editors, which limit off-target events [183]. A sponge-like silica nanoconstruct (SN) harboring Cas9- and BE-RNPs was recently developed for gene editing with low toxicity in mice [184]. Furthermore, improved genome editing via oviductal nucleic acid delivery (i-GONAD) has been widely utilized as an innovative approach to deliver genome editors directly into the murine fallopian tubes via electroporation [185,186,187]. However, insertion of a long DNA fragment using the i-GONAD scheme has been indicated to be less efficient compared to microinjection method [185]. This limitation was overcome with the Easi-CRISPR approach that enables efficient insertion of large DNA donors into genomes via microinjection of the RNPs and donors into mouse zygotes [188,189]. Interestingly, Gu et al. reported that co-injection of monomeric streptavidin-fused Cas9 mRNA, sgRNA, and biotinylated PCR donor DNA into two-cell stage embryos of mice (known as 2C-HR-CRISPR method) could effectively allow large insertions (up to 95%) at the target loci to minimize the undesirable effects on the integration sites [190,191]. In addition, CRISPR-READI is a recently developed method that combines delivery of CRISPR/Cas9 as an RNP via electroporation and donor DNA via an AAV system [192]; this method has been shown to allow successful insertion of large DNA fragments (up to 3.3 kb) into embryos [192]. Collectively, these recent breakthroughs and future advances in gene editing and delivery enable a broad spectrum of human disease models and therapeutic applications, which are expected to facilitate clinical transition and applications of promising programmable nucleases.

## Figures and Tables

**Figure 1 genes-14-00483-f001:**
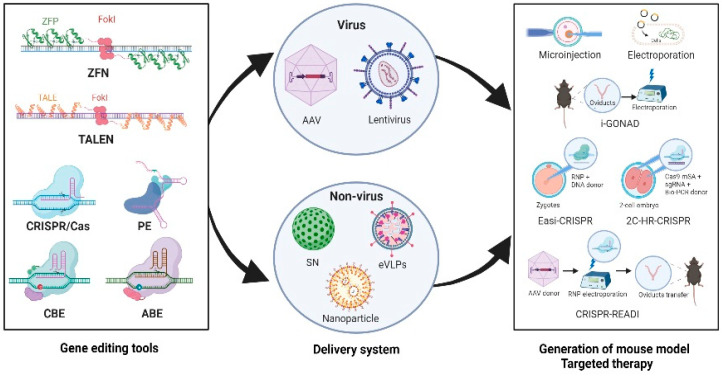
Delivery strategies for gene editing using programmable nucleases in mice. The genome editors, including ZFNs, TALENs, CRISPR/Cas, PE, CBE, and ABE, can be delivered to mice using viral (AAVs and lentiviruses) or nonviral particles (SN, nanoparticles, and eVLPs) via microinjections or electroporation. In addition, the programmable nucleases can be directly delivered to mice using the i-GONAD method without any ex vivo modifications. ZFNs, zinc finger nucleases; TALENs, transcription activator-like effector nucleases; CRISPR, clustered regularly interspaced short palindromic repeats; Cas, CRISPR-associated protein; PE, prime editor; CBE, cytosine base editor; ABE, adenine base editor; AAVs, adeno-associated viral systems, SN, sponge-like silica nanoconstruct; eVLPs, engineered DNA-free virus-like particles; i-GONAD, improved genome editing via oviductal nucleic acid delivery. * This illustration was created with BioRender.com (accessed on 1 February 2023).

## Data Availability

Data availability is not applicable to this article.

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
