# Peer review of "Progress in and Prospects of Genome Editing Tools for Human Disease Model Development and Therapeutic Applications"

_genes, 2023, doi:10.3390/genes14020483_

Round 1

Reviewer 1 Report

 Major comments

The authors aimed to have an overview of the “Progress in and Prospects of Genome Editing Tools for Human Disease Model Development and Therapeutic Applications”. However, the authors used a vast majority of text to list various disease models generated by nucleases delivered by different vehicles. Discussion of the prospects was nearly missed. Moreover, the delivery vehicles should be detailed since it is frequently mentioned and relevant to gene modification efficiency and precision. Furthermore, the shortcomings of the current nuclease systems or delivery approaches should be discussed. Therefore, these aspects should be addressed.

Author Response

Response: Thank you for your insightful comments and critical review. We have added and edited sentences as suggested and indicated on p13-14, lines 495-506, lines 5011-516, lines 519-521, and lines 526-537.

Reviewer 2 Report

This review is well-written and summarized the current status of using genome editing to model human disease. It would be a good paper to learn about the field.

Suggestion for revision:

I suggest include discussion of more technology to increase the efficiency of large fragment knock-in and replacement. Although the original 2013 Cell paper reported ~20% efficiency, it is somewhat hard to reproduce in then transgenic field. A few technologies has been developed since to improve the efficiency include EasiCRISPR (https://genomebiology.biomedcentral.com/articles/10.1186/s13059-017-1220-4, https://www.nature.com/articles/nprot.2017.153), 2C-HR-CRISPR (https://www.nature.com/articles/nbt.4166,https://currentprotocols.onlinelibrary.wiley.com/doi/10.1002/cpmo.67), AAV mediated Knock-in (https://academic.oup.com/nar/article/45/11/e98/3059660?login=true, https://www.sciencedirect.com/science/article/pii/S2211124719307429)

Author Response

Response: Thank you for your perceptive remarks and important review. We have added further discussion of several efficient methods for knocking in large fragments as per your suggestions. Please refer to p13, lines 526-537.

Reviewer 3 Report

The manuscript entitled ‘Progress in and Prospects of Genome Editing ….’ reviews the topic of disease modelling using gene editors that introduce mutations to simulate various disease conditions. Advances with the technologies and their applicability to developing new therapies are discussed. Overall, the manuscript is thorough, written well and should be of interest to a wide readership. However, there are some points that should be addressed before publication:

1        It is not always clear whether discussion of the gene editors addresses their use as inducers of the disease-simulating mutations or whether the gene editors are being used as therapies. The concluding Figure (Figure 1) is a graphical representation of this ambiguity.

2        Some of the important details about the differences in gene editors have been omitted and would assist the reader appreciate the topic better. For example, ZFNs’ efficiency is largely dependent on the influence of neighbouring fingers on the overall efficiency of the engineered nuclease. This is difficult to predict and contributes to ZFNs’ current lack of popularity.

3        Acronyms, e.g. PE and BE, should be spelled out the first time that they are used.

4        Influence of off-target mutation by gene editors on the reliability of disease modelling needs to be provided.

5        The types of mutations that are introduced and how exactly they resemble the natural disease-causing mutations should be described.

6        Similarly, specifics of how gene function is restored after therapeutic gene editing should be provided.

7        The conclusions section deals with delivery alone and does not provide a conclusion that pertains to the whole paper.

8        Figure 1 implies that disease models are generated using viral vectors, while therapeutic gene editor are delivered using non-viral approaches. This is surely not so clear cut, and for example, it is possible that disease-generating gene editors could be delivered with non-viral vectors.

Author Response

Reviewer #3:

The manuscript entitled ‘Progress in and Prospects of Genome Editing ….’ reviews the topic of disease modelling using gene editors that introduce mutations to simulate various disease conditions. Advances with the technologies and their applicability to developing new therapies are discussed. Overall, the manuscript is thorough, written well and should be of interest to a wide readership. However, there are some points that should be addressed before publication:

Response: Thank you for your helpful commentary and critical review. We have added or edited sentences as suggested below.

  1.  It is not always clear whether discussion of the gene editors addresses their use as inducers of the disease-simulating mutations or whether the gene editors are being used as therapies. The concluding Figure (Figure 1) is a graphical representation of this ambiguity.

Response: Thank you. As we described in the manuscript, genome editing tools have been developed to produce various genetically modified mouse models for studying human diseases, by mimicking human diseases via introduction of human pathogenic mutations into their genome. Subsequently, these disease models have been used to evaluate genome editing tools for disease treatment via correction of the pathogenic variants. Therefore, we discussed contributions of the gene editing tools not only for generating genetic mouse models, but also for gene therapies. As per your suggestion, we have edited Figure 1 as indicated on p12.

  1. Some of the important details about the differences in gene editors have been omitted and would assist the reader appreciate the topic better. For example, ZFNs’ efficiency is largely dependent on the influence of neighbouring fingers on the overall efficiency of the engineered nuclease. This is difficult to predict and contributes to ZFNs’ current lack of popularity.

Response: Thank you for your recommendation. We have revised the manuscript as shown on p2, lines 78-80.

  1. Acronyms, e.g. PE and BE, should be spelled out the first time that they are used.

Response: Thank you for your comment. We provided the full name of PE and BE when they are firstly mentioned in the manuscript as indicated on p2, lines 52-53. 

  1. Influence of off-target mutation by gene editors on the reliability of disease modelling needs to be provided.

Response: Thank you for your observation. Off-target issues need to be carefully considered for applications of gene editing tools in generating human disease models and gene therapies. We mentioned off-targets in our manuscript, along with the evaluation of editing efficiency on human disease models. The content can be found on p7, lines 288-295 and p8, lines 305-309 and p11, lines 427-429 and p12, lines 472-474.

  1. The types of mutations that are introduced and how exactly they resemble the natural disease-causing mutations should be described.

Response: Thank you for your suggestion. There are several types of DNA mutations, including deletions, insertions, and base substitutions. We refer the reader to the corresponding section of “3. Generation of Mouse Models of Human Diseases Using Genome Editing Tools (see Table 1 for summary)”.

  1. Similarly, specifics of how gene function is restored after therapeutic gene editing should be provided.

Response: Thank you for your proposition. We refer the reader to the corresponding section of “4. Therapeutic Applications Using Genome Editing Tools (see Table 2 for summary)” in the manuscript.

  1. The conclusions section deals with delivery alone and does not provide a conclusion that pertains to the whole paper.

Response: Thank you. We have added and revised the sentences in the Conclusions, which summarizes the whole manuscript (p13, lines 537-536).

  1. Figure 1 implies that disease models are generated using viral vectors, while therapeutic gene editor are delivered using non-viral approaches. This is surely not so clear cut, and for example, it is possible that disease-generating gene editors could be delivered with non-viral vectors.

Response: Thank you for your critique. We revised Figure 1, which demonstrates the generation of mouse models of human diseases as well as targeted therapy employing both viral and non-viral vectors (p12).

Reviewer 4 Report

Manuscript from Thi and cols. represent a well-designed review of state of the art in genome editing and their potential therapeutic use. The section sequence is precise, and the data available are well ensembled in the redaction. We felt that it could be published. 

Author Response

Response: Thank you for your assessment.

Round 2

Reviewer 1 Report

The manuscript has been improved after revision.